# Fate of Planktonic and Biofilm-Derived *Listeria monocytogenes* on Unwaxed Apples during Air and Controlled Atmosphere Storage

**DOI:** 10.3390/foods12193673

**Published:** 2023-10-06

**Authors:** Natasha Sloniker, Ourania Raftopoulou, Yi Chen, Elliot T. Ryser, Randy Beaudry

**Affiliations:** 1Department of Food Science and Human Nutrition, Michigan State University, East Lansing, MI 48824, USA; 2Department of Plant and Microbial Biology, North Carolina State University, Raleigh, NC 27606, USA; 3Office of Regulatory Science, Center for Food Safety and Applied Nutrition, Food and Drug Administration, College Park, MD 20740, USA; 4Department of Horticulture, Michigan State University, East Lansing, MI 48824, USA

**Keywords:** *Listeria monocytogenes*, apple cultivar, Gala, Honeycrisp, Granny Smith, harvest year, biofilm, air storage, controlled atmosphere storage

## Abstract

Multiple recalls and outbreaks involving *Listeria monocytogenes*-contaminated apples have been linked to the post-harvest packing environment where this pathogen can persist in biofilms. Therefore, this study assessed *L. monocytogenes* survival on apples as affected by harvest year, apple cultivar, storage atmosphere, and growth conditions. Unwaxed Gala, Granny Smith, and Honeycrisp apples were dip-inoculated in an 8-strain *L. monocytogenes* cocktail of planktonic- or biofilm-grown cells (~6.5 log CFU/mL), dried, and then examined for numbers of *L. monocytogenes* during air or controlled atmosphere (CA) (1.5% O_2_, 1.5% CO_2_) storage at 2 °C. After 90 days, air or CA storage yielded similar *L. monocytogenes* survival (*p* > 0.05), regardless of harvest year. Populations gradually decreased with *L. monocytogenes* quantifiable in most samples after 7 months. Apple cultivar significantly impacted *L. monocytogenes* survival (*p* < 0.05) during both harvest years with greater reductions (*p* < 0.05) seen on Gala compared to Granny Smith and Honeycrisp. Biofilm-derived cells survived longer (*p* < 0.05) on *L. monocytogenes*-inoculated Gala and Honeycrisp apples compared to cells grown planktonically. These findings should aid in the development of improved *L. monocytogenes* intervention strategies for apple growers and packers.

## 1. Introduction

*Listeria monocytogenes* (*L. monocytogenes*) is a Gram-positive bacterial foodborne pathogen that causes an estimated 1600 illnesses and 260 deaths annually in the United States alone. The populations most susceptible to listeriosis include newborns and infants, the elderly, pregnant women, and immunocompromised individuals [1]. For these populations, *L. monocytogenes* can spread through the bloodstream and cause septicemia and meningitis. Over 95% of listeriosis infections lead to hospitalization, with a fatality rate of 15–20% [1,2]. In pregnant women, listeriosis is most often associated with fetal loss or death of the newborn infant [1]. The vast majority of listeriosis outbreaks are caused by just three *L. monocytogenes* serotypes, 1/2a, 1/2b, and 4b [2]. Historically, food vehicles associated with *L. monocytogenes* have included unpasteurized dairy products, soft cheeses, and sliced deli meats [3]. In the past decade, *L. monocytogenes* outbreaks have increasingly become associated with foods considered to be “moderate risk” or “low risk,” including stone fruit and caramel apples [3,4].

Apples and apple-derived products have been well documented as vehicles of foodborne illness. In 1997, outbreaks of gastrointestinal illness and hemolytic uremic syndrome caused by *Escherichia coli* O157:H7 were linked to unpasteurized apple juice and apple cider [5]. These outbreaks raised concerns related to pathogen growth and survival on apples. After dip-inoculating apples with *E. coli* O157:H7, Buchanan et al. found the highest populations in the stem and calyx portions, which can entrap contaminants, providing a favorable microenvironment for microbial growth [6]. Conway et al. subsequently reported the growth of *L. monocytogenes* on sliced apples during air and controlled atmosphere at 10 and 20 °C [7]. In 2012 and 2013, the first recalls were issued for *L. monocytogenes*-contaminated diced and sliced apples that were distributed across 36 states. However, at that time, apples had not been identified as a listeriosis risk due to the low acid (pH < 4.0) of the fruit.

New concerns were raised in October 2014 when consumption of *L. monocytogenes*-contaminated caramel apples was traced to 35 cases of listeriosis across 12 states, including 34 hospitalizations and one fatality [4]. Environmental testing along the supply chain identified two *L. monocytogenes* serotype 4b strains in the apple-packing facility that matched two clinical isolates [8,9,10,11]. A second caramel apple outbreak of listeriosis was reported in October 2017, which resulted in three hospitalizations. Although finished product and environmental samples were collected along the supply chain, the only *L. monocytogenes* isolate recovered was a non-outbreak-related strain from the supplier [12]. In December 2017, whole apples and later packaged products containing apple slices were recalled from five states due to *L. monocytogenes* contamination [10]. This recall was followed by two others in 2019 and 2020, involving whole and sliced apples contaminated during processing and packing. Taken together, these recalls and outbreaks reinforce the need to understand better the potential harborage sites in apple packing facilities and the subsequent persistence of *L. monocytogenes* on apples [10].

*L. monocytogenes* is a frequent environmental contaminant of apple packinghouses where it can form biofilms on product contact surfaces, including polishing brushes, roller conveyors, dividers, and brushes under fans and blowers [13,14,15,16,17,18,19,20,21]. Biofilm formation begins with the attachment of cells to a surface, followed by bacterial growth and the production of extracellular polymeric substances. Biofilm formation is impacted by temperature, surface material, and nutrient availability. Once mature, biofilms allow for the flow of nutrients and waste between the cells as well as the release of cells that can contaminate the product or form additional biofilms on other food contact surfaces [16]. Protected from chemical sanitizers and other forms of environmental stress, *L. monocytogenes* can survive within these biofilms for months or years [11,21,22,23,24]. Identical *L. monocytogenes* genotypes have been recovered from packinghouses and patients in other outbreaks involving diced celery (2010), whole cantaloupe (2011), stone fruit (2014), and mung bean sprouts (2014), confirming the long-term persistence of *L. monocytogenes* in packinghouse environments [11]. 

Several studies have assessed the survival of planktonically grown cells of *L. monocytogenes* on different apple cultivars during storage. When Sheng et al. inoculated unwaxed Fuji and Granny Smith apples with a cocktail of *L. monocytogenes* strains to contain either 3.5 or 6.0 log CFU/apple, populations decreased 0.5–3.0 logs during 3 months of storage at 1–10 °C [25]. Limited *L. monocytogenes* reductions have also been reported by others on whole apples after 3 to 5 months of cold storage [25,26,27]. However, none of these studies assessed the viability of biofilm-grown cells of *L. monocytogenes* on apples during long-term storage, multiple harvest seasons, different apple growing regions, or storage room atmosphere.

Given the identified data gaps, this study investigated *L. monocytogenes* survival on apples as impacted by (1) apple cultivar (Gala, Granny Smith, and Honeycrisp), (2) harvest season, (3) storage conditions (air or CA), and (4) the type of inoculum (planktonic- or biofilm-grown cultures) to better mimic the contamination from water or food contact surfaces in apple packinghouses. 

## 2. Materials and Methods

### 2.1. Apples and Storage Conditions

Unwaxed Gala, Granny Smith, and Honeycrisp apples (6–8 cm dia.) were shipped from three major apple-growing regions in the United States (Midwest, Northeast, and Northwest) to Michigan State University during the 2019 and 2020 harvest seasons. Upon arrival, any damaged or under-sized apples were discarded. The remaining apples were stored in 0.93 m^3^ aluminum chambers (Storage Control Systems, Sparta, MI) under air or controlled atmosphere (CA) (1.5% O_2_, 1.5% CO_2_) in a 2 °C cold room and treated within 7 d of arrival with 1 μL/L 1-methylcyclopropene for 24 h to suppress ripening and preserve fruit quality. The chamber atmosphere was regulated by an automated atmosphere control system (ICA 61 Laboratory System: International Controlled Atmosphere Ltd., Paddock Wood, UK), with the temperature monitored continuously. 

### 2.2. Bacterial Strains and Growth Conditions 

Eight *L. monocytogenes* strains belonging to serotypes 1/2a, 1/2b, and 4b came from the laboratory of Dr. Sophia Kathariou at North Carolina State University. The panel included two different strains from the 2014–2015 caramel apple outbreak as well as strains from several other listeriosis outbreaks (Table 1). The strains were barcoded for subsequent metagenomic analysis with unique 30-bp DNA sequences constructed and incorporated into the chromosome as previously described [28]. Plasmid pTZ200.mix—a derivative of the pPL2 plasmid that allows stable incorporation of the barcodes into the chromosome—was extracted from *Escherichia coli* SM10λpir and electroporated into each of the strains [28,29]. Transformants were selected on Brain Heart Infusion (BHI) agar containing chloramphenicol (plasmid marker; 10 μg/mL). Individual colonies were analyzed by PCR and Sanger sequencing to identify uniquely barcoded isolates. For one of the strains, 4b1, barcoding was based on chromosomal *gfp* sequences [30]. Whole genome sequences were obtained for all strains and their parental counterparts. 

The strains were stored at −80 °C in trypticase soy broth containing 0.6% (*w*/*v*) yeast extract (TSBYE, Neogen, Lansing, MI, USA) and 10% (*v*/*v*) glycerol (Sigma-Aldrich, Inc., St. Louis, MO, USA). Working cultures were prepared by streaking the frozen stock culture onto modified tryptic soy agar (mTSAYE) containing 0.6% yeast extract (Neogen), 0.1% (*w*/*v*) esculin, and 0.5% (*w*/*v*) ferric ammonium citrate (Sigma-Aldrich) followed by incubation at 37 °C for 48 h. Each of the barcoded strains was compared to its parental strain for biofilm formation, hemolytic activity, motility on soft agar, and virulence using the *Galleria mellonella* model as previously described [31]. Based on these test results, genetic barcoding did not affect the strain phenotypes.

### 2.3. Preparation of Planktonic and Biofilm-Derived L. monocytogenes Inoculum 

The eight *L. monocytogenes* strains were separately grown (37 °C, 24 h) in TSBYE. After adjusting the OD_600_ values to 0.600–0.650, the cultures were combined in equal volumes, pelleted twice by centrifugation (9000× *g*, 15 min, 4 °C), resuspended in phosphate-buffered saline (PBS) and diluted to ~10^7^ CFU/mL in 1.6 L of deionized (DI) water for apple inoculation. *L. monocytogenes* populations in the inoculum were confirmed by plating appropriate dilutions on mTSAYE and Modified Oxford agar (MOX, Neogen) followed by incubation at 37 °C for 36–48 h. For the biofilm inoculum, the same strains were similarly grown (37 °C, 24 h) in TSBYE, adjusted to the same OD_600_ range, and combined in equal proportions to obtain an 8-strain cocktail. Thereafter, 400 μL of the cocktail was added to each of 80 150 mm-dia. Petri plates followed by 19.6 mL of tryptic soy broth (TSB). After incubation (37 °C, 48 h), the TSB was discarded, and the plates were rinsed twice with 1 mL of PBS. Biofilm cells were harvested from the plates using three sterile PBS-moistened cotton-tipped swabs (Puritan, Guilford, ME, USA) per plate, which were transferred to a sterile 50 mL Corning polypropylene centrifuge tube (ThermoFisher Scientific, Waltham, MA, USA) containing 20 mL of PBS and vortexed for 1 min. After removing the swabs, these 80 biofilm-derived suspensions were combined to obtain the cocktail as described above for planktonic cells and then diluted to ~10^7^ CFU/mL in 1.6 L of deionized (DI) water for apple inoculation.

### 2.4. Apple Inoculation and Storage 

The planktonic and biofilm-derived inoculums (1.6 L) were each added to 14.4 L of DI water in a 56.8 L Nalgene™ Lightweight Graduated Cylindrical Tank (ThermoFisher Scientific) lined with low-density polyethylene bags (ULINE, Pleasant Prairie, WI, USA) to obtain a population of ~6.5 log CFU/mL. 

Groups of ~15 apples were transferred to mesh produce bags (Product Packaging Supplies, Elgin, IL, USA) and initially washed in 5 L of DI water to remove any soil or debris, with the water changed after every six bags. Thereafter, duplicate bags of apples were immersed in 16 L of the *L. monocytogenes* cocktail and continuously agitated for 10 min using a sanitized plastic pole (ULINE, Pleasant Prairie, WI, USA). After draining, the apples were aseptically removed from the bags, dried at room temperature on an aluminum foil-lined shelf while periodically turning to prevent pooling of the inoculum in the stem bowl or calyx, and finally transferred to the chambers described above for storage at 2 °C in air or CA (1.5% O_2_, 1.5% CO_2_).

### 2.5. Sampling and L. monocytogenes Enumeration

Inoculated apples were sampled immediately after air-drying (day 0), weekly during the first month, and then monthly. Two composite samples of three apples each were randomly removed from storage at each time point. After removing the stem bowl and calyx portions with a sterile knife, the remaining skin was removed using an electric apple peeler (Rotato Express, Electric Peeler 093209-006-BLCK, Starfrit, QC, Canada). The stem and calyx portions were added to one Whirl-pak bag (Nasco, Modesto, CA, USA), and the peel was added to a second Whirl-pak bag. The samples were diluted 1:5 in sterile PBS (*w*/*v*) and then homogenized in a stomacher (Stomacher 400 Circulator, Seward, Worthington, UK) for 1 min at 300 rpm. Appropriate PBS dilutions were plated on mTSAYE and MOX with *L. monocytogenes* colonies enumerated after 48 h of incubation at 37 °C. After just 30 days of storage, counts for the peel alone were typically near the limit of detection. Therefore, the counts from the stem/calyx and the peel were combined and expressed as log CFU/apple.

### 2.6. Statistical Analysis

Two independent apple storage trials were conducted in the fall of 2019 and fall of 2020. Two composite samples of three apples each per cultivar and growing region were analyzed at each sampling time. Analysis of covariance (ANCOVA) and post-hoc pairwise comparisons were used to determine statistical significance at *p* ≤ 0.05 using JASP software version 0.14.1 (The JASP Team, Amsterdam, The Netherlands). Response variables analyzed include storage atmosphere, harvest year, apple cultivar, and inoculum type. Box plots were created using RStudio Professional Version 2022.07.0 (RStudio, Inc., Boston, MA, USA). 

## 3. Results

Dip inoculation in the planktonic and biofilm suspensions yielded *L. monocytogenes* populations of 3.22–5.50 and 5.97–6.60 log CFU/apple, respectively, with no significant differences seen between cultivars. Overall, *L. monocytogenes* populations decreased over time; however, the pathogen was still quantifiable at ~2 to 4 log CFU/apple in most samples after 7 months of storage. Some planktonic- and biofilm-inoculated apples still yielded *L. monocytogenes* populations of 4.6 log CFU/apple after 210 days of CA storage.

### 3.1. Storage Atmosphere

After 90 days of storage at 2 °C, no significant difference (*p* > 0.05) in *L. monocytogenes* survival was observed between the apples subjected to air or CA storage, regardless of harvest year (Figure 1)*. L. monocytogenes* populations on Gala and Honeycrisp apples from harvest year 1 decreased ~0 to 3.5 log CFU/apple after 90 days of air and CA storage. However, *L. monocytogenes* was more persistent on Granny Smith apples, decreasing < 1.0 log CFU/apple during 30 to 90 days of air and CA storage. *L. monocytogenes* populations on Gala and Granny Smith apples from harvest year 2 initially decreased ~2 to 2.5 log CFU/apple during 14 to 90 days of air and CA storage, whereas populations on Honeycrisp apples initially decreased ~2.0 log CFU/apple between 30 and 90 days of storage (Figure 1). 

### 3.2. Harvest Year

*L. monocytogenes* persistence was significantly (*p* < 0.05) impacted by harvest year, with greater overall survival observed in harvest year 1 (Figure 2). However, decreased survival of *L. monocytogenes* was observed on Granny Smith apples from harvest year 1 (*p* < 0.05) compared to harvest year 2. After 210 days of storage, *L. monocytogenes* populations on Granny Smith apples decreased <0.64 and 1.40–3.85 log CFU/apple for harvest years 1 and 2, respectively, compared to 1.20–3.15 and 0.93–3.05 log for Honeycrisp apples (Figure 2).

### 3.3. Apple Cultivar

*L. monocytogenes* survival was significantly (*p* < 0.05) impacted by apple cultivar during both harvest years (Figure 2). However, regardless of harvest year, significantly (*p* < 0.05) greater reductions in *L. monocytogenes* were seen for Gala compared to Granny Smith and Honeycrisp apples. After 210 days of storage, *L. monocytogenes* populations on Gala apples decreased from 0.34 to 2.29 and 1.85 to 3.43 log CFU/apple for harvest years 1 and 2, respectively (Figure 2). 

### 3.4. Inoculum Type

Regardless of harvest year, *L. monocytogenes* survival on Granny Smith apples was not significantly (*p* > 0.05) impacted by the type of inoculum (Figure 3 and Figure 4). However, inoculum type did significantly (*p* < 0.05) impact *L. monocytogenes* survival on Gala and Honeycrisp apples. Biofilm-grown cells of *L. monocytogenes* survived significantly (*p* < 0.05) longer than planktonically grown cells on Gala apples for both harvest years and for Honeycrisp apples in harvest year 1 (Figure 3 and Figure 4). After 210 days of storage, *L. monocytogenes* populations on Gala apples inoculated with planktonically grown cells decreased 2.02 and 3.43 log CFU/apple for harvest years 1 and 2, respectively, as compared to 0.56 and 2.18 log CFU/apple for biofilm-grown cells. Numbers of planktonically grown cells on Honeycrisp apples from harvest years 1 and 2 decreased 2.55 and 1.16 log CFU/apple after 210 days of storage, respectively, compared to 0.01 and 2.76 log CFU/apple for biofilm-grown cells.

## 4. Discussion

In this study, the survival of *L. monocytogenes,* which is a facultative anaerobe, was similar between air and CA storage after 90 days (*p* > 0.05), regardless of harvest year. Greater *L. monocytogenes* survival was seen in year 1 compared to year 2 (*p* > 0.05). Apple cultivar significantly impacted *L. monocytogenes* survival (*p* > 0.05) during both harvest years, with greater reductions (*p* > 0.05) seen on Gala compared to Granny Smith and Honeycrisp. Biofilm-derived cells survived longer (*p* > 0.05) on *L. monocytogenes*-inoculated Gala and Honeycrisp apples compared to cells grown planktonically.

Air and CA storage supported similar survival of *L. monocytogenes* on all three apple cultivars. Our findings are supported by Scollard et al., who observed no significant differences (*p* > 0.05) in *L. monocytogenes* survival when a model vegetable system was stored in air and CA (5 and 20% CO_2_) [32]. After harvest, apples are held refrigerated in either air or a controlled atmosphere, which will alter the apple microbiome, decrease the apple respiration rate, and extend shelf life during storage [24,33]. Decreased oxygen levels may also be responsible for enhancing resistance to various environmental stressors [34,35,36].

As previously mentioned, greater *L. monocytogenes* survival was observed in harvest year 1 compared to harvest year 2 (*p* > 0.05). In a study by Bösch et al. that assessed the changing microbiome of apples harvested during 2015–2018, harvest year was one of the two top contributors to both the numbers and diversity of bacteria and fungi, including *Botrytis*, *Monilinia*, *Neofabraea*, and *Penicillium* [37]. For the 2018 harvest year, a significantly higher number of microorganisms were observed, while the 2016 harvest year had a significantly lower number of microorganisms observed compared to the years 2015 and 2017 [37]. Bokulich et al. also reported differences between microbial communities on grapes based on harvest year with net precipitation, maximum temperature, relative humidity, latitude, and longitude most strongly influencing bacterial and fungal growth patterns as well as the taxonomic groups observed [38]. 

In addition to multiple harvest years, multiple apple-growing regions were used in our study to account for variability between regions, orchards, and packinghouse practices. However, the limited number of growers per region precluded any valid comparison between regions. Environmental conditions will differ from year to year. Based on information collected from the National Centers for Environmental Information, the Northwest region (the top apple-producing region in the United States) experienced “much below average” precipitation during harvest year 1 (2019) with “near average” temperatures, compared to year 2 (2020) when “near average” precipitation and “much above average” temperatures were reported [39]. These year-to-year climatic changes, along with the conditions at and near the time of budding, impact both apple yield and quality. 

*L. monocytogenes* survival varies between different apple cultivars. In our study, apple cultivar significantly impacted *L. monocytogenes* survival (*p* > 0.05) during both harvest years with greater reductions (*p* > 0.05) seen on Gala compared to Granny Smith and Honeycrisp. Macarisin et al. also observed significantly lower *L. monocytogenes* survival on unwaxed Red Delicious as compared to Fuji apples after 160 days of simulated commercial storage. Additionally, decreased survival was observed, but not always significant, on unwaxed Red Delicious apples compared to unwaxed Granny Smith apples during 160 days of storage [26]. However, Sheng et al. reported similar survival of *L. monocytogenes* on Granny Smith and Fuji apples during 90 days of air storage [25]. Our findings align with both Sheng et al. and Macarisin et al., with *L. monocytogenes* decreasing 1–2 log on Granny Smith apples after 30 days of refrigerated storage [25,26].

Variations in surface texture and structure between apple cultivars may help explain the observed differences in *L. monocytogenes* attachment and survival. For example, Pietrysiak and Ganjyal found that Gala apples had narrower microcracks (10–100 μm), both in the stem bowl and on the equatorial surface, as compared to Granny Smith (50–150 μm) [40]. Additionally, Gala had smaller microcracks (5 μm) containing internal vertical wax platelets, whereas Granny Smith apples were covered with shallow, wider microcracks (50 μm) and crystalline wax platelets. These microcracks, as well as lenticels and trichomes on the apple surface, can reportedly serve as additional attachment sites for *Listeria innocua* and *Escherichia coli* O157:H7 [16,41]. Decreased *L. monocytogenes* survival on Gala compared to Granny Smith apples in our study reflects these reported differences in surface morphology. 

Multiple studies have shown that pathogen survival in apples is partly dependent on the varying acid, sugar, and polyphenol profiles between cultivars [42]. Jelodarian et al. assessed four apple cultivars for antimicrobial activity against eleven bacterial foodborne pathogens, two cultivars of which exhibited significantly higher antimicrobial activity [43]. Additionally, when Alberto et al. assessed the ability of skin phenolic compounds from Royal Gala and Granny Smith to inhibit *Escherichia coli*, *Staphylococcus aureus*, *Pseudomonas aeruginosa*, *Enterococcus faecalis*, and *Listeria monocytogenes,* Granny Smith apples with higher phenolic content exhibited greater antimicrobial activity against *L. monocytogenes* compared to samples with a lower phenolic content [44]. Since the antioxidant properties of apples also differ between cultivars and harvest seasons [45], the differences between the apple cultivars in our study could also be attributed to a combination of these effects. 

As discussed earlier, biofilm-derived cells survived longer (*p* > 0.05) on *L. monocytogenes*-inoculated Gala and Honeycrisp apples compared to cells grown planktonically. Our findings reflect the enhanced ability of biofilm-grown cells of *L. monocytogenes* to persist longer compared to planktonically-grown cultures. The structure of a biofilm protects *L. monocytogenes* from various environmental stressors, including disinfectants and sanitizers, leading to long-term survival and persistence in such facilities [13]. However, the current research available is still not clear on if there are genetic markers leading to biofilm persistence [46]. Once introduced into apple storage facilities and packinghouses, *L. monocytogenes* can form biofilms in difficult-to-clean locations such as drains, conveyor belts, waxing and packaging equipment, floors, foot baths, and other niches [9,13,16,21,46,47]. While there is robust information to support that *L. monocytogenes* strains are persistent in food processing environments, there is still a need to understand the mechanisms *L. monocytogenes* uses to persist in this environment [46]. 

## 5. Conclusions

The 2015 listeriosis outbreak traced to ice cream confirmed that even low-level contaminated products that do not support *L. monocytogenes* growth can cause life-threatening illness in highly susceptible populations [48]. Our study shows that *L. monocytogenes* can survive on the surface of apples for at least seven months. Therefore, future risk assessments need to account for the survival of apples during long-term refrigerated storage. 

Prevention of *L. monocytogenes* contamination was recently identified as the leading food safety topic of concern among apple packers [49]. Recent environmental sampling for *Listeria* spp. prevalence was done in five Washington state apple packinghouses. It was shown that the food contact surfaces most likely to harbor *Listeria* spp. were polishing brushes, stainless steel dividers, brushes under fans and blowers, and dryer rollers [17]. Therefore, in the future, emphasis needs to be given to eradicating *L. monocytogenes* from these difficult-to-clean niches that are prone to biofilm formation. 

## Figures and Tables

**Figure 1 foods-12-03673-f001:**
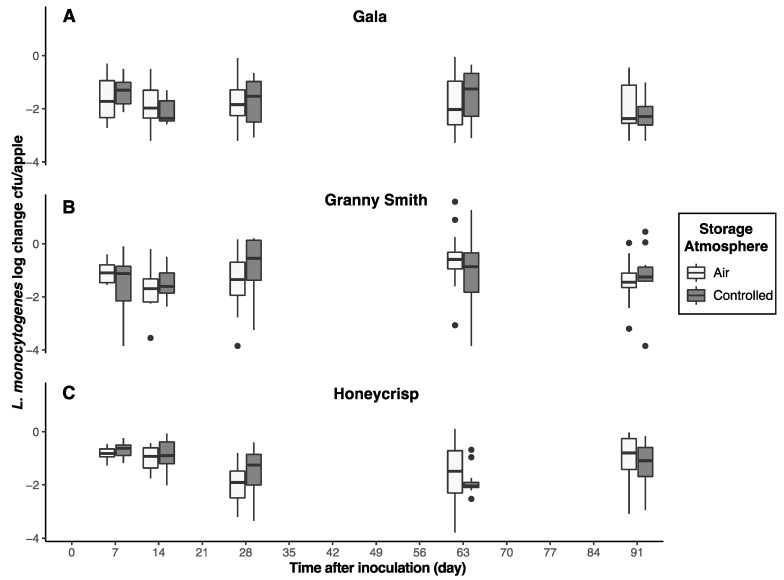
Reductions of *L. monocytogenes* populations on inoculated apples. Reductions were determined as log CFU/apple and shown as boxplots. Unwaxed Gala (**A**), Granny Smith (**B**), and Honeycrisp (**C**) apples were inoculated with cocktails of planktonic cultures and stored in air and controlled atmosphere storage at 2 °C.

**Figure 2 foods-12-03673-f002:**
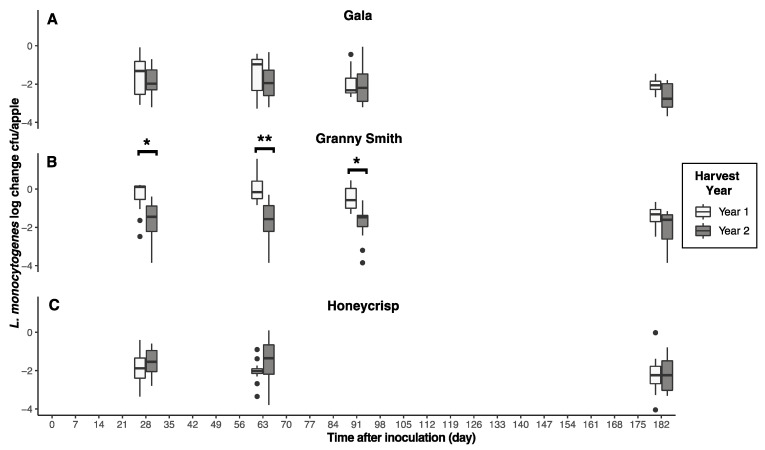
Reductions of *L. monocytogenes* populations on inoculated apples. Reductions were determined as log CFU/apple and shown as boxplots. The asterisk (*) is used to represent a statistically significant result. Unwaxed Gala (**A**), Granny Smith (**B**), and Honeycrisp (**C**) apples from harvest year 1 and harvest year 2 were inoculated with cocktails of planktonic cultures and stored at 2 °C.

**Figure 3 foods-12-03673-f003:**
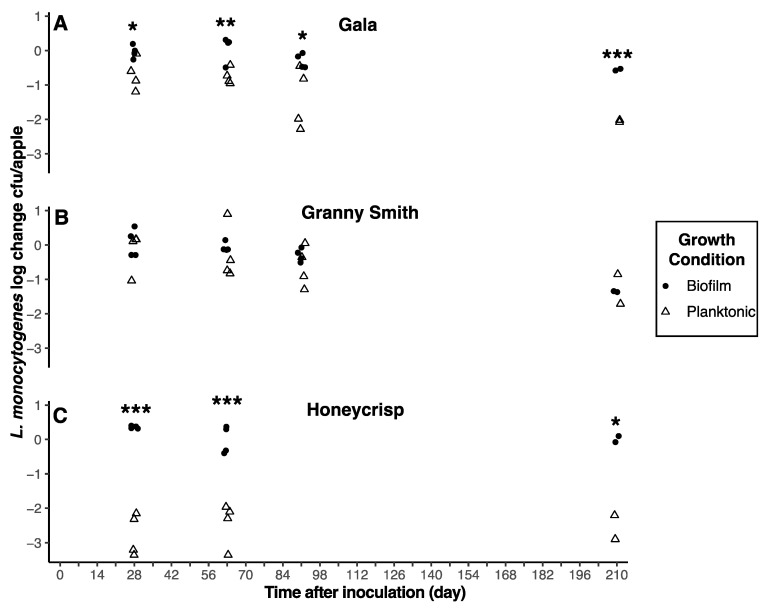
Reductions of *L. monocytogenes* populations on inoculated apples. Reductions were determined as log CFU/apple and shown as scatterplots. The asterisk (*, **, ***) is used to represent a statistically significant (*p* < 0.05, 0.01, 0.001) result. Unwaxed Gala (**A**), Granny Smith (**B**), and Honeycrisp (**C**) apples from harvest year 1 were inoculated with cocktails of biofilm (black circle) and planktonic (white triangle) cultures and stored at 2 °C.

**Figure 4 foods-12-03673-f004:**
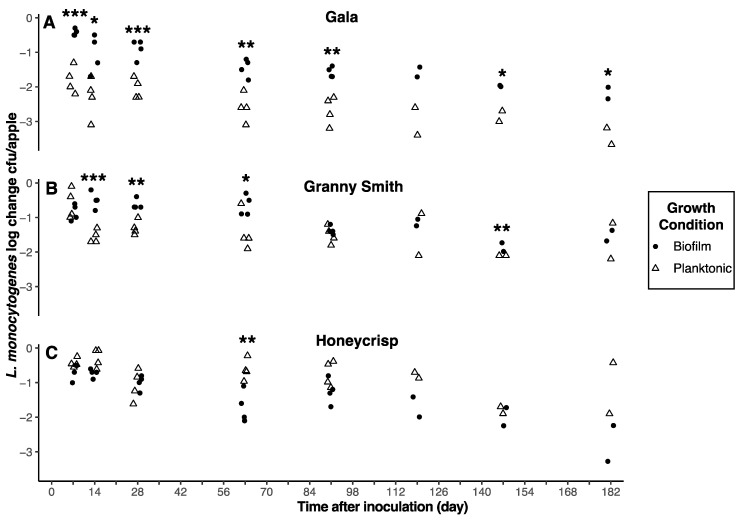
Reductions of *L. monocytogenes* populations on inoculated apples. Reductions were determined as log CFU/apple and shown as scatterplots. The asterisk (*, **, ***) is used to represent a statistically significant (*p* < 0.05, 0.01, 0.001) result. Unwaxed Gala (**A**), Granny Smith (**B**), and Honeycrisp (**C**) apples from harvest year 2 were inoculated with cocktails of biofilm (black circle) and planktonic (white triangle) cultures and stored at 2 °C.

**Table 1 foods-12-03673-t001:** Panel of eight barcoded Listeria monocytogenes strains.

Strain	Serotype	Genotype	Outbreak
4b1-GFP	4b	ST2	Clinical isolate,1962
F2365-2	4b	ST1	California cheese outbreak, 1985
H7858-1	4b	ST6	Hot dog outbreak, 1998–99
2010L-1723-4	1/2a	ST378	Celery outbreak, 2010
CFSAN023957-A10	4bv-1	ST554	Mung bean sprouts outbreak, 2014
2014L-6680-7	4b	ST1	Caramel Apple outbreak, 2014–2015
2014L-6695-5	4b	ST382	Caramel Apple outbreak, 2014–2015
CFSAN073872-6	1/2b	ST581	Apples, 2017

## Data Availability

The data supporting the conclusions from this work are included within the manuscript. No additional datasets were analyzed.

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
