# Peer review of "Fate of Planktonic and Biofilm-Derived *Listeria monocytogenes* on Unwaxed Apples during Air and Controlled Atmosphere Storage"

_foods, 2023, doi:10.3390/foods12193673_

Round 1
Reviewer 1 Report
The article is quite original; some corrections and above all some clarifications are required.
Please see below the comments:
Line 2: The work refers to ….Listeria monocytogenes in apples…. but throughout the text the authors cite “Listeria”very often; It would be appropriate to specify the reference to Listeria monocytogenes as this is the purpose of the work.
Line 18: the word “Listeria” is not necessary
Line 33: please add “(L. monocytogenes)” after Listeria monocytogenes
Line 72: It is not clear whether the authors want to refer to Listeria or to Listeria monocytogenes. Please specify if you referring to L. monocytogenes. The reference to Listeria is too generic in this context. Check this throughout the article (for example, Lines 83-139-177-192-195…and so on)
Line 134: please put “L. monocytogenes” in italics
Line 364: please enter additional information about this reference
Author Response
Reviewer 1
Line 2: The work refers to ….Listeria monocytogenes in apples…. but throughout the text the authors cite “Listeria” very often; It would be appropriate to specify the reference to Listeria monocytogenes as this is the purpose of the work.
Changes were made in all locations where authors previously referred to “Listeria” the text now reads either “Listeria monocytogenes” or “L. monocytogenes”.
Line 18: the word “Listeria” is not necessary
“Listeria” was removed and the sentence now reads, “Therefore, this study assessed L. monocytogenes survival on apples as affected by harvest year, apple cultivar, storage atmosphere, and growth conditions.”
Line 33: please add “(L. monocytogenes)” after Listeria monocytogenes
An edit was made and, the sentence now reads, “Listeria monocytogenes (L. monocytogenes) is a Gram-positive bacterial foodborne pathogen that causes an estimated 1,600 illnesses and 260 deaths annually in the United States alone.”
Line 72: It is not clear whether the authors want to refer to Listeria or to Listeria monocytogenes. Please specify if you referring to L. monocytogenes. The reference to Listeria is too generic in this context. Check this throughout the article (for example, Lines 83-139-177-192-195…and so on)
Changes were made in all locations where authors previously referred to “Listeria” the text now reads either “Listeria monocytogenes” or “L. monocytogenes”.
Line 134: please put “L. monocytogenes” in italics
A change was made and the title now reads:
2.3. Preparation of Planktonic and Biofilm-derived L. monocytogenes Inoculum.
Line 364: please enter additional information about this reference
An edit was made to clear up this statement and include more information from the paper. The conclusion now reads “Recent environmental sampling for Listeria spp. prevalence was done in five Washington state apple packinghouses. It was shown, the food contact surfaces most likely to harbor Listeria spp. were polishing brushes, stainless steel dividers, brushes under fans and blowers, and dryer rollers [17].
Reviewer 2 Report
This study is very well desinged and very important to the field, however, the authors should make thses changes prior consider for publiation.
1- The authors should clearly indicate the scientific hypthesis tested and inlcude in the introduction
2- Why authors chosed this bacteria to be tested on the apples while there are other pathgoen more imporant like E. coli as they indicated in the intoduciton.
3- Why they didn't use different types of food like Ice cream as this bacteria seem to be highly associated with ice cream?
4-The authors should precisely explained why the biofilm forming bacteria survive longer than non- biofilm forming bacteria?
Author Response
Reviewer 2
1- The authors should clearly indicate the scientific hypothesis tested and include in the introduction
In the introduction the identified data gaps and relationships investigated were stated starting on line 95.
2- Why authors chosed this bacteria to be tested on the apples while there are other pathogen more important like E. coli as they indicated in the introduction.
This research was conducted specifically as a response to the 2014-15 caramel apple Listeria monocytogenes outbreak as discussed in lines 58-62.
3- Why they didn't use different types of food like Ice cream as this bacteria seem to be highly associated with ice cream?
This research was conducted specifically as a response to the 2014-15 caramel apple Listeria monocytogenes outbreak as discussed in lines 58-62.
4-The authors should precisely explained why the biofilm forming bacteria survive longer than non- biofilm forming bacteria?
An edit was made in the Discussion to include information on what is known about the mechanisms used by biofilm-derived L. monocytogenes to survive longer compared to planktonic-derived L. monocytogenes.
Reviewer 3 Report
Comments are in manuscript

Author Response
Reviewer 3
Must use scientific names in italics
Changes were made in all locations where authors previously referred to “Listeria” the text now reads either “Listeria monocytogenes” or “L. monocytogenes”.
Line 20 & 205, “Standardize units used” & “Homogenize units throughout”
Changes were made to ensure all units were standardized throughout to read “mL” and “days”.
Line 20, “Insert specific scientific name ‘Listeria monocytogenes or L. monocytogenes’”
Changes were made in all locations where authors previously referred to “Listeria” the text now reads either “Listeria monocytogenes” or “L. monocytogenes”.
Line 57, “do you mean pH”
An edit was made and the sentence now reads, “However, at that time, apples had not been identified as a listeriosis risk due to the low acid (pH < 4.0) of the fruit.”
Line 76, “Check redaction”
An edit was made to the inline citation and the sentence now reads, “Once mature, biofilms allow for the flow of nutrients and waste between the cells as well the release of cells that can contaminate the product or form additional biofilms on other food contact surfaces [46].”
Line 85, “year missing”
Reference List and Citations Style Guide for MDPI Journals instructed in text citations do not include year.
Remove “as described in Materials and Methods” from figures
An edit was made to remove all “as described in Materials and Methods from figure descriptions.
Line 281 & 301 must include the values of the results to compare with the present works
Changes were made to include more information on the significance of the harvest growing years on lines 283-386. of the results for Bösch et al. Values for the results of Macarisin et al. are included on line 307.
Line 344 where do you conclude this from? There is no information to support this conclusion
An edit was made to clear up this statement and include more information from the paper. The conclusion now reads “Recent environmental sampling for Listeria spp. prevalence was done in five Washington state apple packinghouses. It was shown, the food contact surfaces most likely to harbor Listeria spp. were polishing brushes, stainless steel dividers, brushes under fans and blowers, and dryer rollers [17].
Reviewer 4 Report
Dear Authors,
Thank you for submitting this manuscript that has addressed an issue of public concerns. Although L. monocytogenes are largely studied in animal origin food, this manuscript highlighted the presence of the pathogen in plant origin food that can be consumed without heat treatment. The manuscript showed that the pathogen may survive long time storage in a risky number.
The introduction: was adequate to pave the topic for the typical readers, However, I 'd like to see more about biofilm formation by L. monocytogenes (more specific), their ability to survive different conditions and their potential risk with few cells.
The methodology:
For homogenization process by stomacher it is better to refer the quantity by grams this will help to represent the results per gram not per an apple.
results: the difference between apple cultivars wasn't explained adequately. The smoothness of apple skin is one of these factors that authors need to search.
conclusion: was a bit general and directly related to current work. I felt it is part of the introduction to justify the study.
Regards
Author Response
Reviewer 4
The introduction: was adequate to pave the topic for the typical readers, However, I 'd like to see more about biofilm formation by L. monocytogenes (more specific), their ability to survive different conditions and their potential risk with few cells.
An edit was made to the introduction to include more information directly related to biofilm formation and factors that may influence the formation of Listeria monocytogenes biofilms.
The methodology: For homogenization process by stomacher it is better to refer the quantity by grams this will help to represent the results per gram not per an apple.
While referring to the quantity by grams would help with standardizing the results to grams, other work in this area focuses on per apple Listeria monocytogenes counts and this is why the experiment was designed to use per apple counts. Therefore, the weight per gram of samples was not used.
results: the difference between apple cultivars wasn't explained adequately. The smoothness of apple skin is one of these factors that authors need to search.
The differences in texture of the apple skin and content of the apple cuticle are currently being studied in further work.
conclusion: was a bit general and directly related to current work. I felt it is part of the introduction to justify the study.
An edit was made to include more information directly related to current work on Listeria monocytogenes present in apple packinghouses. This information ties in other research currently being done on reducing risk of Listeria monocytogenes contamination.